# Communicating Hierarchical Neural Controllers For Learning Zero-shot Task Generalization

**Junhyuk Oh, Satinder Singh, Honglak Lee**
University of Michigan
Ann Arbor, MI, USA
{junhyuk,baveja,honglak}@umich.edu

**Pushmeet Kohli**
Microsoft Research
Redmond, WA, USA
pkohli@microsoft.com

## Abstract

The ability to generalize from past experience to solve previously unseen tasks is a key research challenge in reinforcement learning (RL). In this paper, we consider RL tasks defined as a sequence of high-level instructions described by natural language and study two types of generalization: to unseen and longer sequences of previously seen instructions, and to sequences where the instructions themselves were previously not seen. We present a novel hierarchical deep RL architecture that consists of two interacting neural controllers: a meta controller that reads instructions and repeatedly communicates subtasks to a subtask controller that in turn learns to perform such subtasks. To generalize better to unseen instructions, we propose a regularizer that encourages to learn subtask embeddings that capture correspondences between similar subtasks. We also propose a new differentiable neural network architecture in the meta controller that learns temporal abstractions which makes learning more stable under delayed reward. Our architecture is evaluated on a stochastic 2D grid world and a 3D visual environment where the agent should execute a list of instructions. We demonstrate that the proposed architecture is able to generalize well over unseen instructions as well as longer lists of instructions.

## 1 Introduction

Humans can often generalize to novel tasks even without any additional learning by leveraging past learning experience. We would like our artificial agents to have similar "zero-shot" generalization capabilities. For example, after learning to solve tasks with instructions such as 'Go to X (or Y)' and 'Pick up Y (or Z)', our agent should be able to infer the underlying goal of new tasks with instructions like 'Go to Z', which requires disentangling the verbs ('Go to/Pick up') and the nouns/objects ('X, Y, Z'). Furthermore, we would like our agents to learn to compose policies to solve novel tasks composed of sequences of seen and unseen instructions. Developing the ability to achieve such generalizations is a key challenge in artificial intelligence and the subfield of reinforcement learning (RL).

In this paper, we study the problem of zero-shot task generalization in RL by introducing the "instruction execution" problem where the agent is required to learn through interaction with its environment how to achieve an overall task specified by a list of high-level instructions (see Figure 1). As motivation for this problem consider a human owner training its new household robot to execute complex tasks specified by natural language text that decompose the task into a sequence of instructions. Given that it is infeasible to explicitly train the robot on all possible instruction-sequences, this problem involves two types of generalizations: to unseen and longer sequences of previously seen instructions, and sequences where the some of the instructions themselves were previously not seen. Of course, the usual RL problem of learning policies through interaction to accomplish the goals of an instruction remains part of the problem as well. We assume that the agent does *not* receive any signal on completing or fail-

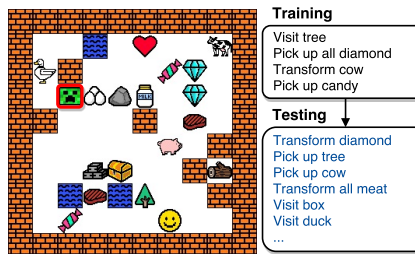

Figure 1: Example of grid-world and instructions. The agent is tasked to execute longer sequences of instructions after trained on short sequences of instructions; in addition previously unseen instructions can be given during evaluation (blue text). The agent can get more rewards if it deals with randomly appearing enemies (red outlined box) regardless of current instructions.

ing to complete individual instructions from the environment/owner and so the informative reward signal is delayed until the end. Furthermore, there can be random events in the environment that require the agent to interrupt whatever it is doing and deviate from the instructions to maintain some background task as described in Figure 1. Altogether this makes for a challenging zero-shot task generalization RL problem.

**Brief Background:** RL tasks composed of sequences of subtasks have been studied before and a number of hierarchical RL approaches designed for them. Typically these have the form of a *meta controller* and a set of lower-level controllers for subtasks (Sutton et al., 1999; Dietterich, 2000; Parr and Russell, 1997). The meta controller is limited to selecting one from a set of lower-level controllers to employ at any time. This makes it impossible for the low-level controller to generalize to new subtasks without training a new low-level controller separately. Much of the previous work also assumes that the overall task is fixed (e.g., *Taxi* domain (Dietterich, 2000; Ghavamzadeh and Mahadevan, 2003)). Transfer learning across multiple compositional tasks has typically been studied in RL formulations in which new tasks are only presented via a new reward function from the environment (Singh, 1991; 1992) and so there is no opportunity for fast model-free generalization. To the best of our knowledge, zero-shot model-free generalization to new or longer tasks as well as unseen tasks has not been well-studied in the RL setting.

**Our Architecture:** This paper presents a hierarchical deep RL architecture (see Figure 2) that consists of two interacting neural controllers: a *meta controller* that repeatedly chooses an instruction and conditioned on the current state of the environment translates it into *subtask-arguments* (details on this in later sections) and communicates those to the *subtask controller* that in turn chooses primitive actions given the subtask. This makes the subtask controller a parameterized option (Sutton et al., 1999) module in which the parameters are the subtask-arguments mentioned above. On top of the subtask controller, the meta controller is trained to select proper subtask-arguments depending on observations from the environment, feedback from the subtask controller about termination, and the task instructions. In order to generalize over unseen instructions, we propose analogy-making regularization (discussed in Section 4.1) which encourages to learn subtask embeddings that capture correspondences between similar subtasks. In addition, we propose a new differentiable neural architecture in the meta controller that implicitly learns temporal abstractions so that it can operate at a larger time-scale and update the subtask-arguments to the subtask controller only when needed.

**Our Results:** We developed a 2D grid world environment where the agent can interact with many objects as illustrated in Figure 1 based on MazeBase (Sukhbaatar et al., 2015) (see Section 6.1 for details). The empirical results show that the meta-controller's ability to learn temporal abstractions and a form of analogy-making regularization were all key in allowing our hierarchical architecture to generalize in a zero-shot fashion to unseen tasks. We also demonstrated that the same architecture can also generalize to unseen and longer instructions in a 3D visual environment.

## 2    RELATED WORK

**Hierarchical Reinforcement Learning.**    In addition to hierarchical RL described in Section 1, there is a line of work on portable options for solving sequential tasks (Konidaris et al., 2012; Konidaris and Barto, 2007). They proposed agent-space options that can be re-used to deal with new problems. However, the optimal sequence of options (e.g., picking up a key followed by opening a door) is fixed throughout training and evaluation in their problem. On the other hand, the agent is required to perform new sequences of tasks depending on given instructions in our work. Our work is also closely related to Programmable HAM (PHAM) (Andre and Russell, 2000; 2002) in that PHAM is designed to execute a given program. However, the program explicitly specifies the policy in PHAM which effectively reduces state-action space. In contrast, a list of instructions is a partial description of the task in our work, which means that the policy is not forced to follow the instructions but to use them as a guide to maximize its reward. For example, interrupt conditions need be manually specified by the program in PHAM, while they are not specified in the instructions but should be learned by the agent in our framework.

Hierarhical RL has been recently combined with deep learning. Kulkarni et al. (2016) proposed hierarchical Deep Q-Learning and demonstrated improved exploration in a challenging Atari game. Tessler et al. (2016) proposed a similar architecture that allows the high-level controller to choose primitive actions directly. Bacon and Precup (2015) proposed *option-critic* architecture which learns options without any domain knowledge and demonstrated that it can learn distinct options in Atari

games. Vezhnevets et al. (2016) proposed a deep architecture that automatically learns macro-actions. Unlike these recent works that aim to solve a single task, the goal of our work is to build a multi-task policy that can generalize over many different sequences of tasks.

**Zero-shot Task Generalization and Parameterized Option.**   There has been only a few studies that aim to generalize over new tasks in a zero-shot fashion (i.e., without additional learning). da Silva et al. (2012) proposed the concept of parameterized skill which maps a set of task descriptions to policies. Similarly, Isele et al. (2016) proposed a method for zero-shot task generalization which uses task descriptors to predict the parameter of the policy and proposed a coupled dictionary learning with sparsity constraints to enable zero-shot learning. Schaul et al. (2015) proposed *universal value function approximators* (UVFA) that learn a value function given a state and goal pair and showed that their framework can generalize over unseen goals. Borsa et al. (2016) proposed to learn a representation of state and action shared across different tasks. However, the proposed approach lacks the ability to solve new tasks in a zero-shot way. Our subtask controller implements the idea of parameterized skill or universal option. Unlike the previous works, however, we propose to build a high-level controller (meta controller) on top of the subtask controller to deal with sequential tasks.

**Instruction Execution.**   There has been a line of work for building agents that can execute natural language instructions: Tellex et al. (2011; 2014) for robotics and MacMahon et al. (2006); Chen and Mooney (2011); Mei et al. (2015) for a simulated environment. However, these approaches focus on natural language understanding to map instructions to a sequence of actions or *groundings* in a supervised setting. In contrast, we focus on generalization to different sequences of instructions without any supervision for language understanding or for actions. Branavan et al. (2009) also tackle a similar problem of mapping from natural language instructions to a sequence of actions through RL. However, the agent is given a single sentence at a time from the environment, while the agent has to deal with a full list of instructions in our problem. In addition, they do not discuss how to deal with unseen instructions which is the main focus of our paper.

## 3 OVERVIEW

**Goal.**   We aim to learn a multi-task policy which is a mapping $\pi : \mathcal{S} \times \mathcal{M} \to \mathcal{A}$ where $\mathcal{S}$ is a set of states (or observations), $\mathcal{M}$ is a set of lists of instructions, and $\mathcal{A}$ is a set of primitive actions. More importantly, since $\mathcal{M}$ can be arbitrary large, our goal is to find an optimal policy $\pi^*$ on a very small set of lists of instructions $\mathcal{M}' \subset \mathcal{M}$ such that $\pi^*$ is also optimal in the entire set of lists of instructions $\mathcal{M}$.

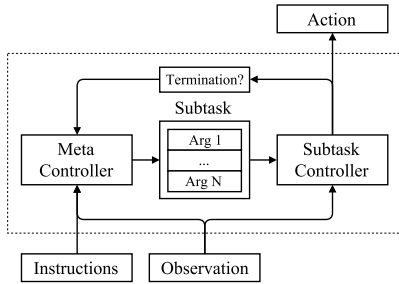

Figure 2: Overview of our architecture

**Hierarchical Structure and Communication Protocol.** As illustrated in Figure 2, the proposed architecture consists of a meta controller which selects a subtask and a subtask controller which executes the given subtask. The subtask is further decomposed into several arguments. More specifically, a space of subtasks $\mathcal{G}$ is defined using the Cartesian product of their arguments $\mathcal{G}^{(1)} \times \cdots \times \mathcal{G}^{(n)}$, where $\mathcal{G}^{(i)}$ is a set of the $i$-th arguments (e.g., $\mathcal{G} = \{\text{Visit}, \text{Pick up}\} \times \{\text{A}, \text{B}\}$). In addition, the subtask controller provides a useful information to meta controller by giving a terminal signal for the given subtask. This communication protocol allows each controller to not only focus on their own independent roles but also communicate with each other to learn a complex closed-loop policy.

**Subtask Controller.**   The subtask controller is a mapping $\mathcal{S} \times \mathcal{G} \to \mathcal{A} \times \mathcal{B}$ which maps a state and a subtask to an action and a termination signal ($\mathcal{B} = \{0, 1\}$) indicating whether the subtask is finished or not. The subtask controller is trained prior to training the meta controller. The main challenge for the subtask controller is that only a subset of subtasks ($\mathcal{U} \subset \mathcal{G}$) is observed during training, and it should be able to generalize over unseen subtasks without experiencing them. Section 4 describes how to construct the subtask architecture parameterized by a neural network and discusses how to generalize over unseen subtasks.

**Meta Controller.**   The meta controller is a mapping $\mathcal{S} \times \mathcal{M} \times \mathcal{G} \times \mathcal{B} \to \mathcal{G}$ which decides a subtask from a state, a list of instructions, a subtask that is currently being executed, and whether the subtask is finished as input. Thus, the meta controller should understand natural language instructions and pass proper subtask arguments to the subtask controller.

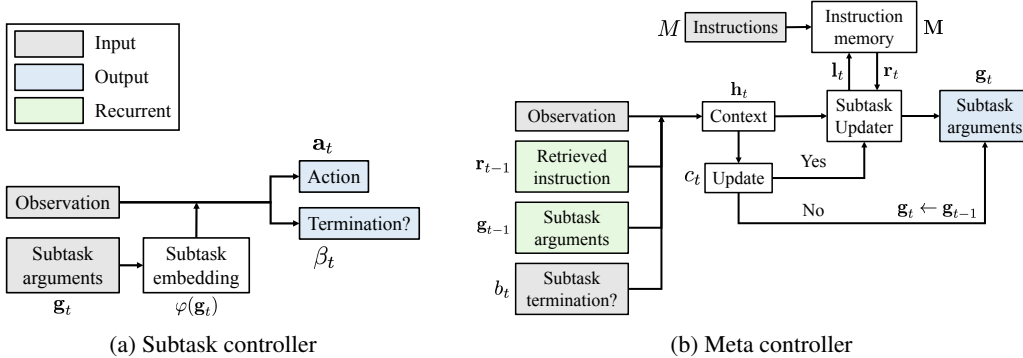

Figure 3: Proposed neural network architectures. See text for details.

It is important to note that natural language instructions are not directly subtasks; indeed there is not a one-to-one correspondence between instructions and subtask-arguments. This is due to a number of important reasons. First, instructions such as 'Pick up all X' are executed by repeatedly solving a subtask [Pick up, X]. Second, the meta controller sometimes needs to interrupt ongoing subtasks and replace them with other subtasks that are not relevant to the instruction because of the background task based on the stochastic events as described in Figure 1.

Another challenge for the meta controller is that it should deal with partial observability induced by the list of instructions. This is because the agent is *not* given which instruction to execute at each time-step from the environment but given just a full list of instructions. Thus, the meta controller should remember how many instructions it has executed and decide when to move to the next instruction. Section 5.1 describes how to construct a memory-based neural network to deal with this challenge.

Finally, it is desirable for the meta controller to operate in a larger time-scale due to the fact that a subtask does not change frequently once it is chosen. We describe a novel way to implicitly learn such a temporal scale of the meta controller through neural networks in Section 5.2.

## 4  SUBTASK CONTROLLER

Given an observation $\mathbf{s}_t \in \mathcal{S}$ and subtask arguments $\mathbf{g} = \left[g^{(1)}, ..., g^{(n)}\right] \in \mathcal{G}$, the subtask controller is defined as the following functions:

$$\text{Policy: } \pi_\phi(\mathbf{a}_t|\mathbf{s}_t, \mathbf{g}) \qquad\qquad \text{Termination: } \beta_\phi(b_t|\mathbf{s}_t, \mathbf{g}) = P_\phi(\mathbf{s}_t \in \mathcal{T}_\mathbf{g})$$

where $\pi_\phi$ is the policy optimized for the subtask. $\beta_\phi$ is a termination function which is a probability that the state is terminal or not for given a subtask. $\mathcal{T}_g$ is the set of terminal states. The subtask controller is parameterized by $\phi$ which is represented by a neural network as illustrated in Figure 3a. The network learns a representation of the subtask $\varphi(\mathbf{g})$, and it is used to condition the entire network through multiplicative interactions as suggested by Memisevic and Hinton (2010); Lei Ba et al. (2015); Bertinetto et al. (2016). Further details are described in Appendix F.

### 4.1  LEARNING TO GENERALIZE BY ANALOGY-MAKING

When learning a non-linear subtask embedding from arguments, $\varphi(\mathbf{g})$, it is desirable for the network to learn prior knowledge about the relationship between different subtask arguments in order to infer the goal of unseen configurations of arguments. To this end, we propose a novel analogy-making regularizer inspired by Reed et al. (2015); Hadsell et al. (2006); Reed et al. (2014). The main idea is to learn correspondences between subtasks. For example, if target objects and 'Visit/Pick up' tasks are independent, we can enforce [Visit, X] : [Visit, Y] :: [Pick up, X] : [Pick up, Y] for any X and Y in the embedding space so that the agent learns to perform [Pick up, Y] as it performs [Pick up, X] and vice versa.

More specifically, we define several constraints as follows:

$$\|\varphi(\mathbf{g}_A) - \varphi(\mathbf{g}_B) - \varphi(\mathbf{g}_C) + \varphi(\mathbf{g}_D)\| \approx 0 \qquad \text{if } \mathbf{g}_A : \mathbf{g}_B :: \mathbf{g}_C : \mathbf{g}_D \qquad (1)$$

$$\|\varphi(\mathbf{g}_A) - \varphi(\mathbf{g}_B) - (\mathbf{g}_C) + \varphi(\mathbf{g}_D)\| \geq \tau_{dis} \qquad \text{if } \mathbf{g}_A : \mathbf{g}_B \neq \mathbf{g}_C : \mathbf{g}_D \qquad (2)$$

$$\|\varphi(\mathbf{g}_A) - \varphi(\mathbf{g}_B)\| \geq \tau_{diff} \qquad \text{if } \mathbf{g}_A \neq \mathbf{g}_B \qquad (3)$$

where $\mathbf{g}_k = \left[ g_k^{(1)}, g_k^{(2)}, ..., g_k^{(n)} \right] \in \mathcal{G}$ are subtask arguments. Eq. (1) represents the analogy-making relationship, while Eq. (2) and and Eq. (3) prevent trivial solutions. To satisfy the above constraints, we propose the following objective functions based on contrastive loss (Hadsell et al., 2006):

$$\mathcal{L}_{sim} = \mathbb{E}_{(\mathbf{g}_A, \mathbf{g}_B, \mathbf{g}_C, \mathbf{g}_D) \sim \mathcal{G}_{sim}} \left[ \| \varphi(\mathbf{g}_A) - \varphi(\mathbf{g}_B) - (\mathbf{g}_C) + \varphi(\mathbf{g}_D) \|^2 \right] \tag{4}$$

$$\mathcal{L}_{dis} = \mathbb{E}_{(\mathbf{g}_A, \mathbf{g}_B, \mathbf{g}_C, \mathbf{g}_D) \sim \mathcal{G}_{dis}} \left[ \max\left(0, \tau_{dis} - \| \varphi(\mathbf{g}_A) - \varphi(\mathbf{g}_B) - (\mathbf{g}_C) + \varphi(\mathbf{g}_D) \| \right)^2 \right] \tag{5}$$

$$\mathcal{L}_{diff} = \mathbb{E}_{(\mathbf{g}_A, \mathbf{g}_B) \sim \mathcal{G}_{diff}} \left[ \max\left(0, \tau_{diff} - \| \varphi(\mathbf{g}_A) - \varphi(\mathbf{g}_B) \| \right)^2 \right] \tag{6}$$

where $\mathcal{G}_{sim}, \mathcal{G}_{dis}, \mathcal{G}_{diff}$ consist of subtask arguments satisfying conditions in Eq. (1), Eq. (2) and Eq. (3) respectively. $\tau_{dis}, \tau_{diff}$ are threshold distances (hyperparameters). The final analogy-making regularizer is the weighted sum of the above three objectives.

**Analogies Under Non-independence.** Although we use analogy-making regularizer so that all configurations of subtasks arguments are valid and independent from each other throughout the main experiment, our analogy-making regularizer can also be used to inject prior knowledge so that the agent generalizes to unseen subtasks in a specific way. For example, if some objects should be handled in a different way given the same subtask, we can apply analogy-making regularizer so that Eq. 1 is satisfied only between the same type of objects. This is further discussed in Appendix B.

## 4.2 TRAINING

The subtask controller is trained on a subset of subtasks ($\mathcal{U} \subset \mathcal{G}$) by directly providing subtask arguments. The policy of the subtask controller is trained through the actor-critic method (Konda and Tsitsiklis, 1999) with generalized advantage estimation (GAE) (Schulman et al., 2015). We also found that pre-training the subtask controller through *policy distillation* (Rusu et al., 2015; Parisotto et al., 2015) gives slightly better results. The idea of policy distillation is to train separate policies for each subtask and use them to provide action labels to train the subtask controller. Throughout training, the subtask controller is also made to predict whether the current state is terminal or not through a binary classification objective, and analogy-making regularizer is applied to the subtask embedding separately. The full details of the learning objectives are described in Appendix E.1.

## 5 META CONTROLLER

The role of the meta controller is to decide subtask arguments $\mathbf{g}_t \in \mathcal{G}$ from an observation $\mathbf{s}_t \in \mathcal{S}$, a list of instructions $M \in \mathcal{M}$, the previously selected subtask $\mathbf{g}_{t-1}$, and its termination signal ($b \sim \beta_\phi$) from the subtask controller. Section 5.1 describes the overall architecture of the meta controller for dealing with the partial observability induced by the list of instructions as discussed in Section 3. We describe a novel way to learn the time-scale of the meta controller so that it can implicitly operate in a large time-scale in Section 5.2.

## 5.1 ARCHITECTURE

In order to keep track of its progress on instruction execution, the meta controller maintains its internal state by computing a *context* vector (described in Section 5.1.1) and by focusing on one instruction at a time from the list of instructions $M$ (described in Section 5.1.2). The entire architecture is illustrated in Figure 3b and further details are described in Appendix F.

### 5.1.1 CONTEXT

Given the sentence embedding $\mathbf{r}_{t-1}$ retrieved at the previous time-step from the instructions (described in Section 5.1.2), the previously selected subtask $\mathbf{g}_{t-1}$, and the subtask termination $b_t \sim \beta_\phi\left(b_t | \mathbf{s}_t, \mathbf{g}_{t-1}\right)$, the meta controller computes the context vector ($\mathbf{h}_t$) through a neural network:

$$\mathbf{h}_t = f_\theta\left(\mathbf{s}_t, \mathbf{r}_{t-1}, \mathbf{g}_{t-1}, b_t\right)$$

where $f_\theta$ is a neural network parameterized by $\theta$. Intuitively, $\mathbf{g}_{t-1}$ and $b_t$ provide information about which subtask was being solved by the subtask controller and whether it has been finished or not. Note that the subtask does not necessarily match with the retrieved instruction ($\mathbf{r}_{t-1}$), e.g., when the agent is dealing with the background task. By combining all the information, $\mathbf{h}_t$ encodes the spatio-temporal context which is used to determine which instruction to solve and the next subtask.

### 5.1.2 SUBTASK UPDATER

The meta controller has a *subtask updater* that constructs a memory structure from the list of instructions, retrieves an instruction by maintaining a pointer into the memory structure, and computes the subtask arguments.

**Instruction Memory.** Given instructions as a list of sentences $M = (m_1, m_2, ..., m_K)$, where each sentence consists of a list of words, $m_i = (w_1, ..., w_{|m_i|})$, the 'subtask updater constructs memory blocks $\mathbf{M} \in \mathbb{R}^{E \times K}$, where each column is $E$-dimensional embedding of a sentence. The subtask module maintains a *memory pointer* defined over memory locations, $\mathbf{p}_t \in \mathbb{R}^K$, which is used for instruction retrieval. Memory construction and retrieval is formally described as:

$$\text{Memory: } \mathbf{M} = [\varphi^w(m_1), \varphi^w(m_2), ..., \varphi^w(m_K)] \qquad \text{Retrieval: } \mathbf{r}_t = \mathbf{M}\mathbf{p}_t.$$

Here $\varphi^w(m_i) \in \mathbb{R}^E$ is the embedding of the $i$-th sentence (e.g., Bag-of-words). The memory pointer $\mathbf{p}_t$ is a non-negative vector which sums up to 1. $\mathbf{r}_t \in \mathbb{R}^E$ is the retrieved sentence embedding which is used for computing the subtask-arguments. Intuitively, if the memory pointer is a one-hot vector, $\mathbf{r}_t$ indicates a single instruction from the whole list of instructions. The meta controller should learn to manage $\mathbf{p}_t$ so that it can focus on the correct instruction at each time-step, which is further described below.

**Location-based Memory Addressing.** Since instructions should be executed sequentially, we use a location-based memory addressing mechanism (Zaremba and Sutskever, 2015; Graves et al., 2014) to manage the memory pointer. Specifically, the subtask updater shifts the memory pointer by $[-1, 1]$ as:

$$\mathbf{p}_t = \mathbf{l}_t * \mathbf{p}_{t-1} \text{ where } \mathbf{l}_t \sim \text{Softmax}\left(\varphi^{shift}(\mathbf{h}_t)\right) \tag{7}$$

where $*$ is a convolution operator, and $\varphi^{shift}$ is a multi-layer perceptron (MLP). $\mathbf{l}_t \in \mathbb{R}^3$ is an internal action that shifts the memory pointer ($\mathbf{p}_t$) by either -1, 0, or +1. This mechanism is illustrated in Figure 9b.

**Subtask Arguments.** The subtask updater takes the context ($\mathbf{h}_t$), updates the memory pointer ($\mathbf{p}_t$), retrieves a sentence embedding ($\mathbf{r}_t$), and finally computes subtask-arguments as follows:

$$\pi_\theta\left(\mathbf{g}_t | \mathbf{h}_t, \mathbf{r}_t\right) = \prod_i \pi_\theta\left(g_t^{(i)} | \mathbf{h}_t, \mathbf{r}_t\right) \text{ where } \pi_\theta\left(g_t^{(i)} | \mathbf{h}_t, \mathbf{r}_t\right) \propto \exp\left(\varphi_i^{goal}(\mathbf{h}_t, \mathbf{r}_t)\right)$$

where $\varphi_i^{goal}$ is an MLP for the $i$-th subtask argument.

### 5.2 DIFFERENTIABLE TEMPORAL ABSTRACTIONS

Although the subtask updater can update the memory pointer and compute correct subtask-arguments in principle, making a decision at every time-step can be inefficient because subtasks do not change very frequently. Instead, having temporally-extended actions can be useful for dealing with delayed reward by operating at a larger time-scale (Sutton et al., 1999). Although one could use the termination signal of the subtask controller to define the temporal scale of the meta controller, this approach would result in an open-loop policy that is not able to interrupt ongoing subtasks, which is necessary to deal with stochastic events.

---

**Algorithm 1** Subtask update (Hard)

**Input:** $\mathbf{h}_t, \mathbf{p}_{t-1}, \mathbf{r}_{t-1}, \mathbf{g}_{t-1}$
**Output:** $\mathbf{p}_t, \mathbf{r}_t, \mathbf{g}_t$
$c_t \sim \sigma\left(\varphi^{update}(\mathbf{h}_t)\right)$
**if** $c_t = 1$ **then** ▷ Update
 $\mathbf{l}_t \sim \text{Softmax}\left(\varphi^{shift}(\mathbf{h}_t)\right)$
 $\mathbf{p}_t \leftarrow \mathbf{l}_t * \mathbf{p}_{t-1}$ ▷ Shift
 $\mathbf{r}_t \leftarrow \mathbf{M}^\top \mathbf{p}_t$ ▷ Retrieve
 $\mathbf{g}_t \sim \pi_\theta\left(\mathbf{g}_t | \mathbf{h}_t, \mathbf{r}_t\right)$ ▷ Subtask
**else**
 $\mathbf{p}_t \leftarrow \mathbf{p}_{t-1}, \mathbf{r}_t \leftarrow \mathbf{r}_{t-1}, \mathbf{g}_t \leftarrow \mathbf{g}_{t-1}$
**end if**

---

To address this challenge, we introduce an internal binary action $c_t$ which decides whether to update the subtask updater or not. This action is defined as: $c_t \sim \sigma\left(\varphi^{update}(\mathbf{h}_t)\right)$. If $c_t = 1$, the subtask updater updates the memory pointer, retrieves an instruction, and updates the subtask arguments. Otherwise, the meta controller continues communicating the current subtask arguments without involving the subtask updater. During training of the update decision, we use L1 regularization on the probability of update to penalize frequent updates as in Vezhnevets et al. (2016). The entire scheme is described in Algorithm 1.

However, the update decision introduces a non-differentiable variable which is known to be difficult to optimize in practice. Thus, we propose a differentiable relaxation of the update decision. The key idea is to take the weighted sum of both 'update' and 'no update' scenarios. This idea is described in Algorithm 2. We found that training the meta controller using Algorithm 2 followed by fine-tuning using Algorithm 1 is crucial for training the meta controller. Note that Algorithm 2 reduces to Algorithm 1 if we sample $c_t$ and $\mathbf{l}_t$ instead of taking the weighted sum, which justifies our initialization trick.

---

**Algorithm 2** Subtask update (Soft)

**Input:** $\mathbf{h}_t, \mathbf{p}_{t-1}, \mathbf{r}_{t-1}, \mathbf{g}_{t-1}$
**Output:** $\mathbf{p}_t, \mathbf{r}_t, \mathbf{g}_t$
$c_t \leftarrow \sigma\left(\varphi^{update}\left(\mathbf{h}_t\right)\right)$
$\mathbf{l}_t \leftarrow \text{Softmax}\left(\varphi^{shift}\left(\mathbf{h}_t\right)\right)$
$\tilde{\mathbf{p}}_t \leftarrow \mathbf{l}_t * \mathbf{p}_{t-1}$
$\tilde{\mathbf{r}}_t \leftarrow \mathbf{M}^\top \tilde{\mathbf{p}}_t$
$\mathbf{p}_t \leftarrow c_t \tilde{\mathbf{p}}_t + (1 - c_t)\mathbf{p}_{t-1}$
$\mathbf{r}_t \leftarrow c_t \tilde{\mathbf{r}}_t + (1 - c_t)\mathbf{r}_{t-1}$
$g_t^{(i)} \sim c_t \pi_\theta\left(g_t^{(i)} | \mathbf{h}_t, \tilde{\mathbf{r}}_t\right)$
$\qquad + (1 - c_t)g_{t-1}^{(i)} \forall i$

---

### 5.3 TRAINING

The meta controller is trained on a training set of lists of instructions. Actor-critic method is used to update the parameters of the meta controller, while a pre-trained subtask controller is given and fixed. Since the meta controller also learns a subtask embedding $\varphi(\mathbf{g}_{t-1})$ and has to deal with unseen subtasks during evaluation, we applied analogy-making regularization to its embedding. More details of the objective functions are provided in Appendix E.

## 6 EXPERIMENTS AND RESULTS

Our experiments were designed to explore the following hypotheses: our proposed hierarchical architecture will generalize better than a non-hierarchical controller, that analogy-making regularization and learning temporal abstractions in the meta controller will both separately be beneficial for task generalization. We are also interested in understanding the qualitative properties of our agent's behavior. The demo videos are available at the following website: `https://sites.google.com/a/umich.edu/junhyuk-oh/task-generalization`.

### 6.1 EXPERIMENTAL SETTING

**Environment.** We developed a 2D grid world based on MazeBase (Sukhbaatar et al., 2015) where the agent can interact with many objects as illustrated in Figure 1. Unlike the original MazeBase, an observation is represented as a binary 3D tensor: $\mathbf{x}_t \in \mathbb{R}^{18 \times 10 \times 10}$ where 18 is the number of object types and $10 \times 10$ is the size of the grid world. Each channel is a binary mask indicating the presence of each object type. There are agent, blocks, water, and 15 types of objects with which the agent can interact (see Appendix D), and all of them are randomly placed for each episode.

The agent has 13 primitive actions: *No-operation*, *Move* (North/South/West/East, referred to as "NSWE"), *Pick up* (NSWE), and *Transform* (NSWE). *Move* actions move the agent by one cell in the specified direction. *Pick up* actions remove the adjacent object in the corresponding relative position, and depending on the object type *Transform* actions either remove it or transform it to another object.

The agent receives a time penalty $(-0.1)$ for each time-step. Water cells act as obstacles which give $-0.3$ when the agent visits them. The agent receives $+1$ reward when it finishes all instructions in the correct order. Throughout the episode, an enemy randomly appears, moves, and disappears after 10 steps. Transforming an enemy gives $+0.9$ reward. More details are described in the appendix D.

**Subtasks and Instructions.** The subtask space is defined as the Cartesian product of two arguments: $\mathcal{G} = \{\text{Visit}, \text{Pick up}, \text{Transform}\} \times \{X_1, X_2, ..., X_{15}\}$ where $X_i$ is an object type. The agent should be on the same cell of the target object to finish 'Visit' task. For 'Pick up' and 'Transform' tasks, the agent should perform the corresponding primitive action to the target object. If there are multiple target objects in the world, the agent can perform the action to any of the target objects.

The instructions are represented as a sequence of sentences, each of which is one of the following: **Visit X**, **Pick up X**, **Transform X**, **Pick up all X**, and **Transform all X** where 'X' is the target object type. While the first three instructions require the agent to perform the corresponding subtask, the last two instructions require the agent to repeat the same subtask until the target objects completely disappear from the world.

**Task Split.** Among 45 subtasks in $\mathcal{G}$, only 30 subtasks are presented to the subtask controller during training. 3 subtasks from the training subtasks and 3 subtasks from the unseen subtasks

| Agent | Train | | | Unseen | | |
|---|---|---|---|---|---|---|
| | Reward | Success | Accuracy | Reward | Success | Accuracy |
| w/o Analogy | **0.56** | **99.9%** | **100.0%** | -1.88 | 60.8% | 49.6% |
| w/ Analogy | **0.56** | **99.9%** | **100.0%** | **0.55** | **99.8%** | **99.6%** |

Table 1: Performance of subtask controller. 'Analogy' indicates analogy-making regularization. 'Accuracy' represents termination prediction accuracy. We assume a termination prediction is correct only if predictions are correct throughout the whole episode.

were selected as the validation set to pick the best-performing subtask controller. For training the meta controller, we created four sets of sequences of instructions: training, validation, and two test sets. The training tasks consist of sequences of up to 4 instructions sampled from the set of training instructions. The validation set consists of sequences of 7 instructions with small overlaps with the training instructions and unseen instructions. The two test sets consist of 20 seen and unseen instructions respectively. More details of the task split are described in the appendix D.

**Flat Controller.** To understand the advantage of using the communicating hierarchical structure of our controllers, we trained a flat controller which is almost identical to the meta controller architecture except that it directly chooses primitive actions without using the subtask controller. Details of the flat controller architecture are described in the appendix F. The flat controller is pre-trained on the training set of subtasks. To be specific, we removed the instruction memory and fed a single instruction as an additional input (i.e., $\mathbf{r}_t$ is fixed throughout the episode). We found that the flat controller could not learn any reasonable policy without this pre-training step which requires modification of the architecture based on domain knowledge. After pre-training, we fine-tuned the flat controller with the instruction memory on lists of instructions. Note that the flat controller is also capable of executing instructions as well as dealing with random events in principle.

## 6.2 TRAINING DETAILS

The subtask controller consists of 3 convolution layers and 2 fully-connected layers and takes the last 2 observations concatenated through channels as input. Each subtask argument ($g^{(i)}$) is linearly transformed and multiplied with each other to compute the joint subtask embedding. This is further linearly transformed into the weight of the first convolution layer, and the weight of the first fully-connected layer. The meta controller takes the current observation as input and has 2 convolution layers and 2 fully-connected layers where the parameters of the first convolution layer and the first fully-connected layer are predicted by the joint embedding of $\mathbf{r}_{t-1}, \varphi(\mathbf{g}_{t-1})$, and $b_t$.

We implemented synchronous actor-critic with 16 CPU threads based on MazeBase (Sukhbaatar et al., 2015), each of which samples a mini-batch of episodes ($K$) in parallel. The parameters are updated after $16 \times K$ episodes. The details of architectures and hyperparameters are described in the appendix F.

**Curriculum Learning via a Forgiving World.** We conducted curriculum training by changing the size of the grid world, the density of objects, and the number of instructions according to the agent's success rate. In addition, we trained the soft-architectures on an easier *forgiving* environment which generates target objects whenever they do not exist. Crucially, this allows the agent to recover from past mistakes in which it removed needed target objects. The soft-architectures are fine-tuned on the original (and far more unforgiving) environment which does not regenerate target objects in the middle of the episode. Training directly in the original environment without first training in the forgiving environment leads to too much failure at executing the task and the agent does not learn successfuly. Finally, the hard-architectures are initialized by the soft-architectures and further fine-tuned on the original environment.

## 6.3 EVALUATION OF SUBTASK CONTROLLER

To see how well the subtask controller performs separately from the meta controller, we evaluated it on the training set of subtasks and unseen subtasks in Table 1. It is shown that analogy-making regularization is crucial for generalization to unseen subtasks. This result suggests that analogy-making regularization plays an important role in learning the relationship between different subtasks and enabling generalization to unseen subtasks.

In addition, we observed that the subtask controller learned a non-trivial policy by exploiting causal relationships. For example, when [Pick up, egg] is given as the subtask arguments, but a duck is very close to the agent, it learned to transform the duck and pick up the resulting egg because

| | Train | Test #1 | Test #2 | Test #3 | Test #4 |
|---|---|---|---|---|---|
| Set of instructions | Seen | Seen | Unseen | Seen w/o all | Unseen w/o all |
| Num of instructions | 4 | 20 | 20 | 20 | 20 |
| **Forgiving** — Shortest Path | -1.56 (99.6%) | -11.94 (99.1%) | | -9.62 (99.1%) | |
| Near-Optimal | -0.96 (99.6%) | -9.99 (99.1%) | | -8.19 (99.1%) | |
| Flat | -1.64 (85.8%) | -14.53 (65.9%) | -17.25 (23.7%) | -12.38 (60.4%) | -14.18 (16.7%) |
| Hierarchical-TA-Analogy | **-1.05** (92.4%) | **-11.06** (86.2%) | **-13.69** (51.2%) | **-8.54** (91.9%) | **-9.91** (75.2%) |
| **Original** — Shortest Path | -1.62 (99.7%) | -11.94 (99.4%) | | -8.72 (99.6%) | |
| Near-Optimal | -1.34 (99.5%) | -10.30 (99.3%) | | -7.62 (99.4%) | |
| Flat | -2.38 (76.0%) | -18.83 (0.1%) | -18.92 (0.0%) | -15.09 (0.0%) | -15.17 (0.0%) |
| Hierarchical | -2.04 (72.8%) | -16.85 (16.6%) | -17.66 (6.9%) | -10.99 (49.4%) | -11.40 (47.4%) |
| Hierarchical-Analogy | -1.74 (81.0%) | -15.89 (28.0%) | -17.23 (11.3%) | -10.11 (61.8%) | -10.66 (57.7%) |
| Hierarchical-TA | -1.38 (92.6%) | -12.96 (62.9%) | -17.19 (13.0%) | -9.11 (74.4%) | -10.37 (61.2%) |
| Hierarchical-TA-Analogy | **-1.26** (95.5%) | **-11.30** (81.3%) | **-14.75** (40.3%) | **-8.24** (85.5%) | **-9.51** (73.9%) |

Table 2: Performance of meta controller. Each column corresponds to different evaluation sets of instructions, while each row corresponds to different configurations of our architecture and the flat controller. Test #3 and Test #4 do not include 'Transform/Pick up all X' instructions. 'TA' indicates the meta controller with temporal abstraction. Each entry in the table represents reward with success rate in parentheses averaged over 10-best runs among 20 independent runs. 'Shortest Path' is a hand-designed policy which executes instructions optimally based on the shortest path but ignores enemies. 'Near-Optimal' is a near-optimal policy that executes instructions based the shortest path and transforms enemies when they are close to the agent. 'Forgiving' rows show the result from the forgiving environment used for curriculum learning where target objects are regenerated whenever they do not exist in the world.

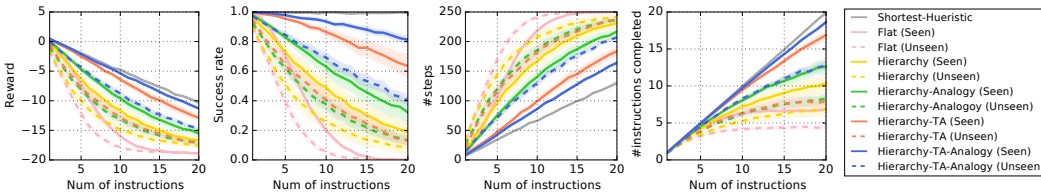

Figure 4: Performance per number of instructions. From left to right, the plots show reward, success rate, the number of steps, and the average number of instructions completed respectively. Solid and dashed curves show the performances on seen instructions and unseen instructions respectively.

transforming the duck transforms it to an egg in our environment. More analysis of the subtask controller and the effect of analogy-making regularization is discussed in the appendix A and B.

## 6.4 EVALUATION OF META CONTROLLER

We evaluated the meta controller separately from the subtask controller by providing the best-performing subtask controller during training and evaluation. The results are summarized in Table 2 and Figure 4. Note that there is a discrepancy between reward and success rate, because success rate is measured only based on the instruction execution, while reward takes into account the background task (i.e., handling randomly appearing enemy) as well as the instruction execution.

**Overall performance.** Table 2 shows that our hierarchical agent with temporal abstraction and analogy-making regularization, denoted Hierarchical-TA-Analogy in the table, can handle 20 seen instructions (Test #1) and 20 unseen instructions (Test #2) correctly with reasonably high success rates. In addition, that agent learned to deal with enemies whenever they appear, and thus it outperforms the 'Shortest Path' policy which is near-optimal in executing instructions while ignoring enemies. We further investigated how the number of instructions affects the performance in Figure 4. Although the performance is degraded as the number of instructions increases, our architecture finishes 18 out of 20 seen instructions and 12 out of 20 unseen instructions on average. These results show that our agent is able to generalize to longer compositions of instructions as well as unseen instructions by just learning to solve short sequences of a subset of instructions.

**Flat vs. Hierarchy.** All our hierarchical controllers outperform the flat controller both on the training tasks and longer/unseen instructions (see Table 2). We observed that the flat controller learned a sub-optimal policy which assumes that 'Transform/Pick up X' instructions are identical to 'Transform/Pick up all X' instructions. In other words, it always transforms or picks up all existing targets. Although this simple strategy is a reasonable sub-optimal policy because such wrong actions are not explicitly penalized in our environment other than through the accumulating penalty per-

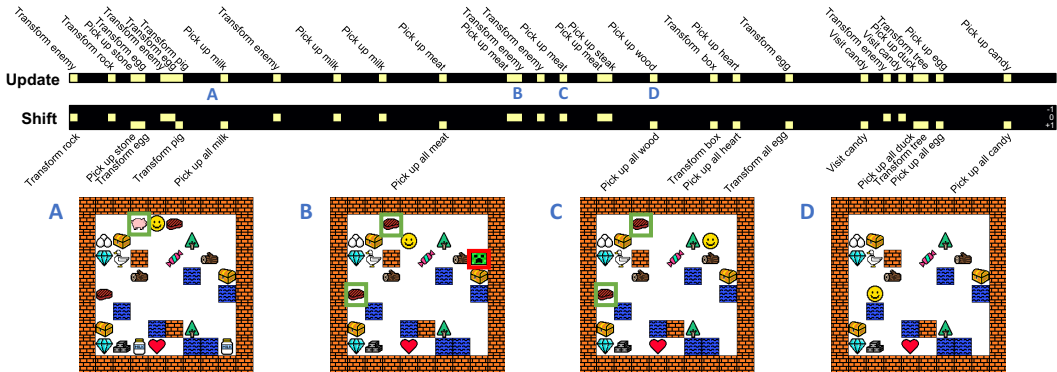

Figure 5: Analysis of the learned policy. 'Update' shows our agent's internal update decision. 'Shift' shows our agent's memory-shift decision which is either -1, 0, or +1 from top to bottom. The bottom text shows the instruction indicated by the memory pointer, while the top text shows the subtask chosen by the meta controller. (A) the agent transforms the pig given 'Transform Pig' instruction and decides to update the subtask (Update is true) and move to the next instruction. (B) an enemy (red) appears while the agent is executing 'Pick up all meat' instruction (green boxes for meat). The agent changes the subtask to [Transform, enemy]. (C) the agent successfully transforms the enemy and sets the subtask to [Pick up, meat] to resume executing the instruction. (D) the agent picks up the last meat in the world, moves the memory pointer to the next instruction, and sets a new subtask according to the next instruction.

time-step, it often unnecessarily removes objects that can be potentially target objects in the future instructions. This is why the flat controller performs reasonably well on the short sequences of instructions (training) where such cases are rare and on the forgiving environment where target objects are restored whenever needed. But, it completely fails on longer instructions in the original environment because the entire task becomes unsolvable when target objects are removed in error. This implies that the flat controller struggles with detecting when a subtask is finished precisely, whereas our hierarchical controllers can easily detect when a subtask is done, because the subtask controller in our communicating architecture provides a termination signal to the meta controller.

In addition, the flat controller tends to ignore enemies, while the hierarchical controllers try to deal with enemies whenever they exist by changing the subtask-arguments communicated by the meta controller to the subtask controller, which is a better strategy to maximize the reward. The flat controller instead has to use primitive actions to deal with both instructions and enemies. This implies that our communicating hierarchical controllers have more advantages for context switching between different sources of tasks (i.e., executing instructions and dealing with enemies).

Finally, we observed that the flat controller often makes many mistakes on unseen instructions (e.g., transform X given 'Visit X' as instruction). In contrast, the hierarchical controllers do not make such mistakes as the subtask controller generalizes well to unseen instructions as discussed in Section 6.3.

**Effect of Analogy-making.** Table 2 shows that analogy-making significantly improves generalization performance especially on Test #2 (Hierarchical-Analogy outperforms Hierarchical, and Hierarchical-TA-Analogy outperforms Hierarchical-TA). This implies that given an unseen target object for the 'Transform/Pick up all' instruction, the meta controller without analogy-making tends to fail to check if the target object exists or not. On the other hand, there is almost no improvement by using analogy-making on Test #3 and Test #4 where there are no 'all' instruction. This is because the meta controller can simply rely on the subtask termination ($b_t$) given by the subtask controller to check if the current instruction is finished for non-'all' instructions, and the subtask controller (trained with analogy-making) successfully generalizes to unseen subtasks and provides accurate termination signals to the meta controller. The empirical results showing that analogy-making consistently improves generalization performance in both non-analogy-making controllers suggests that analogy-making is crucial for generalization to unseen tasks.

**Effect of Temporal Abstraction.** To see the effect of temporal abstractions, we trained a baseline that updates the memory pointer and the subtask at every time-step which is shown as 'Hierarchical' and 'Hierarchical-Analogy' in Table 2. It turns out that the agent without temporal abstractions performs much worse both on the training tasks and testing tasks. We hypothesize that temporal credit assignment becomes easier with temporal abstractions because the subtask updater (described in Section 5.1.2) can operate at a larger time-scale by decoupling the update decision from the

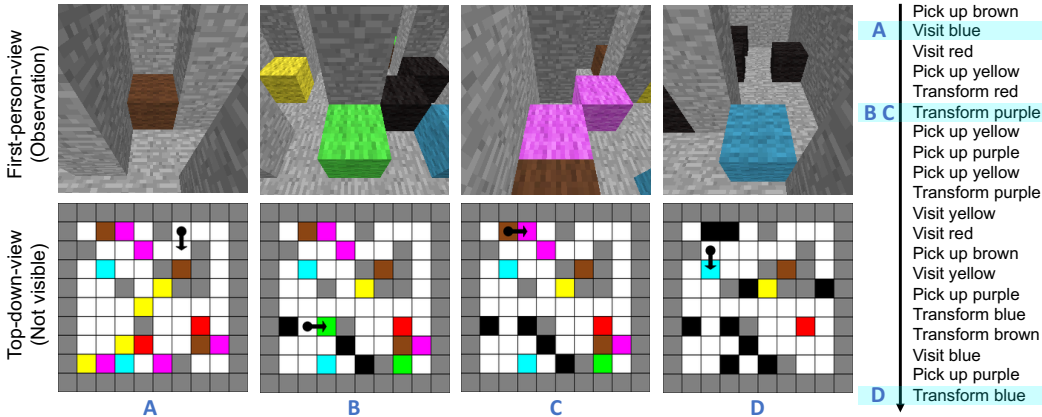

Figure 6: Learned policy in 3D environment. The agent observes 'First-person-view' images, while 'Top-down-view' is not available to the agent. The right text shows the list of instructions. (A) The agent cannot see the target block (blue) at this point due to the partially observable nature of the environment and the randomness of the topology. The agent learned to explore the map to find the target block. (B) Although the current instruction is 'Transform purple', the agent decides to transform the green block because transforming a green block gives a large positive reward (stochastic event). (C) After dealing with the stochastic event, the agent resumes executing the instruction (Traansform purple). (D) The agent finishes the whole list of instructions.

| | Train | Test #1 | Test #2 |
|---|---|---|---|
| Set of instructions | Seen | Seen | Unseen |
| Num of instructions | 4 | 20 | 20 |
| Flat | -1.87 (92.2%) | -22.35 (68.7%) | -39.24 (0.0%) |
| Ours | **-1.41** (95.0%) | **-15.60** (92.2%) | **-17.80** (84.3%) |

Table 3: Performance on 3D environment.

subtask selection. In particular, given 'all' instructions, the agent should repeat the same subtask while not changing the memory pointer for a long time and the reward is even more delayed. This can possibly confuse the subtask updater without temporal abstractions because it should make the same decision for the entire time-steps of such instructions. In contrast, the subtask updater with temporal abstractions can get a direct feedback from the long-term future, since one decision made by the subtask updater results in multiple primitive actions. We conjecture that this is why the agents learn more stably with temporal abstractions under delayed reward.

**Analysis of The Learned Policy.** We visualized our agent's behavior on a task with a long list of instructions in Figure 5. We observed that our meta controller learned to communicate the correct subtask-arguments to the subtask controller and learned to move precisely to the next instruction by shifting the memory pointer whenever the instruction is finished. More interestingly, whenever an enemy appears, our meta controller immediately changes the subtask to [Transform, enemy] regardless of the instruction and resumes executing the instruction after dealing with the enemy. Throughout the background task and the 'all' instructions, the meta controller keeps the memory pointer unchanged as illustrated in (B-D) in the figure. In addition, the agent learned to update the memory pointer and the subtask-argument almost only when it is needed, which provides the subtask updater with temporally-extended actions. This is not only computationally efficient but also useful for learning a better policy as discussed above.

### 6.5 EVALUATION IN 3D VISUAL ENVIRONMENT

We developed a similar set of tasks in Minecraft environment based on Oh et al. (2016) as shown in Figure 6. In this environment, the agent can observe only the first-person-view images, which naturally involves partial observability. In this environment, even executing a simple instruction (e.g., Visit X) requires the agent to explore the topology to find the target.

An observation is represented as a $64 \times 64$ RGB image ($\mathbf{x}_t \in \mathbb{R}^{3 \times 64 \times 64}$). There are 7 different types of colored blocks: red, blue, green, yellow, brown, purple, and black which correspond to different types of objects in the grid world experiment. Like 2D grid world environment, the topology of

walls and the colored blocks are randomly generated for every episode. A wall not only acts as an obstacle but also occludes the objects behind it as shown in Figure 6, which makes the task more challenging.

The agent has 9 actions: *Look* (Left/Right/Up/Down), *Move* (Forward/Backward), *Pick up*, *Transform*, and *No operation*. *Look left/right* actions change the yaw of the agent by 90 degree, while *Look up/down* actions change the pitch of the agent by 45 degree. *Move forward/backward* actions move the agent by one block according to the agent's looking direction. *Pick up* removes the block in front of the agent, and *Transform* changes the block in front of the agent to the black-colored block.

We used the same reward function used in the 2D grid world experiment. In addition, a green block randomly appears and transforming a green block gives +0.9 reward regardless of instructions, which acts as a stochastic event. Each instruction is one of the following: Visit X, Pick up X, and Transform X where 'X' is the target color. We excluded 'all' instructions in this environment because we found that solving 'all' instructions given a limited amount of time is extremely challenging even for humans due to the partial observability.

We used almost the same architectures used in the 2D grid world experiment except that a long short-term memory (Hochreiter and Schmidhuber, 1997) is added on top of the final convolution layer both in the subtask controller and the meta controller, as it is one of the simplest ways to deal with partial observability (Hausknecht and Stone, 2015; Mnih et al., 2016; Oh et al., 2016). We followed the same training scheme used in the 2D grid world experiment.

Table 3 shows that our proposed architecture significantly outperforms the flat controller baseline especially on the test sets of instructions. We observed that the flat controller tends to struggle with detecting when an instruction is finished and completely fails on unseen instructions, while our architecture performs well on unseen and longer instructions. As shown in Figure 6, our architecture learned to find the target blocks, detect when an instruction is finished, and deal with the stochastic event. This result demonstrates that the proposed approach can also be applied to a more complex visual environment.

## 7 CONCLUSION

In this paper, we explored zero-shot task generalization in RL with a new problem where the agent is required to execute a sequence of instructions and to generalize over longer sequences of (un-seen) instructions without additional learning. To solve the problem, we presented a hierarchical deep RL architecture in which a meta controller learns a closed-loop policy of subtask-argument communications to a subtask controller which executes the given subtask and communicates its accomplishment back to the meta controller. Our architecture not only generalizes to unseen tasks after training but also deals with random events relevant to a background task. In addition, we proposed several techniques that led to improvements in both training and generalization performance. First, analogy-making regularization turned out to be crucial for generalization to unseen subtasks. Second, learning temporal abstractions improved the performance by making the subtask updater operate at a larger time-scale. One interesting line of future work would be to define and solve richer task instructions such as conditional statements (i.e., IF-THEN-ELSE) and loop instructions (i.e., collect 3 target objects). Moreover, end-to-end training of the whole hierarchy and discovering the subtask decomposition would be important future work.

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

## A    LEARNED VALUE FUNCTION VISUALIZATION

We visualized the value function learned by the critic network of the subtask controller in Figure 7. As expected from its generalization performance, our subtask controller trained with analogy-making regularization learned high values around the target objects given unseen subtasks.

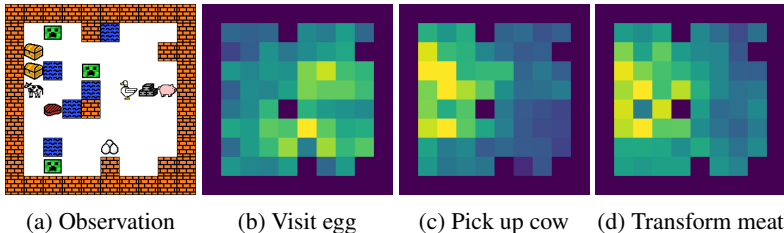

(a) Observation     (b) Visit egg     (c) Pick up cow     (d) Transform meat

Figure 7: Value function visualization given unseen subtasks. (b-d) visualizes learned values for each position of the agent in a grid world (a). The agent estimates high values around the target object in the world.

## B    INJECTING PRIOR KNOWLEDGE THROUGH ANALOGY-MAKING

As discussed in Section 4.1, the assumption that subtask arguments are independent from each other may not hold in the real-world. In this experiment, we simulate such a case by introducing a new subtask, **Interact with X**, which requires the agent to perform either 'Pick up' or 'Transform' depending on object type. We divided objects into two groups: Group A should be picked up given 'Interact with' subtasks, while Group B should be transformed.

Although it is impossible to generalize to unseen target objects in this setting, humans can still easily generalize if someone teaches them by saying 'Interact with X as you do with Y' where X is unseen but Y is seen. We claim that our analogy-making regularizer can be used to mimic such a generalization scenario. To empirically verify this, we presented only a subset of target objects to the agent for 'Interact with X' subtasks during training, while the agent observes all target objects for the original subtasks (i.e., Visit, Pick up, Transform). In the meantime, we applied analogy-making regularization only within Group A and Group B separately.

The result in Table 4 shows that the subtask controller successfully generalizes to unseen target objects by picking up target objects for Group A and transforming them for Group B. This result suggests that analogy-making can also be used as a tool for injecting (minimal but sufficient) prior knowledge so that the agent generalizes to unseen tasks in a specific way without having any experience on such tasks.

| Agent | Train | | | Unseen | | |
|---|---|---|---|---|---|---|
| | Reward | Success | Accuracy | Reward | Success | Accuracy |
| w/o Analogy | **0.55** | **99.9%** | **99.9%** | -3.23 | 42.1% | 44.1% |
| w/ Analogy | **0.55** | **99.9%** | **99.9%** | **0.55** | **99.8%** | **99.6%** |

Table 4: Injecting prior knowledge through analogy-making. 'Unseen' column shows performances on unseen 'Interact with X' subtasks. 'Reward', 'Success', and 'Accuracy' represent reward, success rate, and termination prediction accuracy, respectively.

## C    HARD VS. SOFT

Table 5 compares the hard-architecture described in Algorithm 1 against the soft-architecture described in Algorithm 2. It is shown that the hard-architecture outperforms the soft-architecture on unseen and longer instructions, while the soft-architecture performs as well as the hard-architecture on the training set of instructions. This is because the soft-architecture tends to diffuse the memory pointer over memory locations when it is not certain about its decision. In fact, there is no advantage of using the soft-architecture in this problem because the agent should focus on one instruction at a time. Nevertheless, training the soft-architecture is very important because it is used to initialize the hard-architecture. Otherwise, we observed that it is difficult to train the hard-architecture from scratch because its non-differentiable operations make optimization difficult.

|  | Train | Test #1 | Test #2 | Test #3 | Test #4 |
|---|---|---|---|---|---|
| Set of instructions | Seen | Seen | Unseen | Seen w/o all | Unseen w/o all |
| Num of instructions | 4 | 20 | 20 | 20 | 20 |
| Soft | **-1.27** (95.1%) | **-11.80** (74.8%) | -16.24 (22.0%) | **-7.93** (88.9%) | **-9.53** (72.6%) |
| Hard | **-1.26** (95.5%) | **-11.30** (81.3%) | **-14.75** (40.3%) | **-8.24** (85.5%) | **-9.51** (73.9%) |

Table 5: Comparison of the hard-architecture and the soft-architecture.

# D  ENVIRONMENT AND TASKS

**Environment.** The types of objects are illustrated in Figure 8. 'Transform' action either transforms an object to a different object or removes it depending on its type as described in Figure 8.

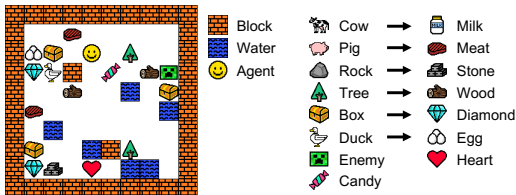

Figure 8: Example of grid-world with object specification. The arrows represent the outcome of object transformation. Objects without arrows disappear when transformed. The agent is not allowed to go through blocks and gets a penalty for going through water.

**Task Split.** For training and evaluating the subtask controller, we constructed a training set of subtasks for training and a validation set for selecting the best-performing agent. These sets are also used to pre-train the flat controller. The details of the sets of subtasks are described in Table 6. For training the meta controller, we constructed a training set of instructions and a validation set of instructions described in Table 7. By sampling instructions from such sets of instructions, we generated different sets of sequences of instructions for training, validation and evaluation in Table 8.

|  | Train (Seen) | | | Validation | | |
|---|---|---|---|---|---|---|
|  | Visit | Pick up | Transform | Visit | Pick up | Transform |
| Cow | ✓ |  | ✓ | ✓ |  |  |
| Enemy | ✓ |  | ✓ |  |  |  |
| Pig | ✓ |  | ✓ |  |  | ✓ |
| Rock | ✓ |  | ✓ |  | ✓ |  |
| Tree | ✓ |  | ✓ |  |  |  |
| Candy | ✓ | ✓ |  |  | ✓ |  |
| Diamond | ✓ | ✓ |  |  |  |  |
| Milk | ✓ | ✓ |  |  |  |  |
| Pork | ✓ | ✓ |  |  |  |  |
| Wood | ✓ | ✓ |  |  |  | ✓ |
| Box |  | ✓ | ✓ | ✓ |  |  |
| Duck |  | ✓ | ✓ |  |  |  |
| Egg |  | ✓ | ✓ |  |  |  |
| Heart |  | ✓ | ✓ |  |  |  |
| Stone |  | ✓ | ✓ |  |  |  |

Table 6: Set of subtasks. 'Train (Seen)' shows subtasks used to train the subtask controller. The other unchecked subtasks are used as the unseen set of subtasks for evaluation.

|  | Train (Seen) | | | | | Validation | | | | |
|---|---|---|---|---|---|---|---|---|---|---|
|  | Visit | Pick up | Transform | Pick up all | Transform all | Visit | Pick up | Transform | Pick up all | Transform all |
| Cow | ✓ |  | ✓ |  | ✓ | ✓ |  |  |  |  |
| Pig | ✓ |  | ✓ |  | ✓ |  |  | ✓ |  |  |
| Rock | ✓ |  | ✓ |  | ✓ |  | ✓ |  | ✓ |  |
| Tree | ✓ |  | ✓ |  | ✓ |  |  |  |  |  |
| Candy | ✓ | ✓ |  | ✓ |  |  | ✓ |  |  |  |
| Diamond | ✓ | ✓ |  | ✓ |  |  |  |  | ✓ |  |
| Milk | ✓ | ✓ |  | ✓ |  |  |  |  |  | ✓ |
| Pork | ✓ | ✓ |  | ✓ |  |  |  |  |  |  |
| Wood | ✓ | ✓ |  | ✓ |  |  |  | ✓ |  |  |
| Box |  | ✓ | ✓ | ✓ | ✓ | ✓ |  |  |  |  |
| Duck |  | ✓ | ✓ | ✓ | ✓ |  |  |  |  | ✓ |
| Egg |  | ✓ | ✓ | ✓ | ✓ |  |  |  |  |  |
| Heart |  | ✓ | ✓ | ✓ | ✓ |  |  |  |  |  |
| Stone |  | ✓ | ✓ | ✓ | ✓ |  |  |  |  |  |

Table 7: Set of instructions. 'Train' and 'Validation' columns show the set of instructions used for training and validation. The unseen set of instructions are defined as the unchecked instructions in 'Train' column.

|  | Train | Validation | Test #1 | Test #2 | Test #3 | Test #4 |
|---|---|---|---|---|---|---|
| Set of instructions | Seen | Unseen | Seen | Unseen | Seen w/o all | Unseen w/o all |
| Max num of instructions | 4 | 7 | 20 | 20 | 20 | 20 |
| Max steps | 60 | 90 | 250 | 250 | 200 | 200 |

Table 8: Task split.

# E    DETAILS OF LEARNING OBJECTIVES

## E.1    SUBTASK CONTROLLER

The subtask controller is first trained through *policy distillation* (Rusu et al., 2015; Parisotto et al., 2015) and fine-tuned using actor-critic method (Konda and Tsitsiklis, 1999) with generalized advantage estimation (GAE) (Schulman et al., 2015). The subtask controller is also trained to predict whether the current state is terminal or not through binary classification objective.

The idea of policy distillation is to first train separate teacher policies ($\pi_T^g(a|s)$) for each subtask ($g$) through reinforcement learning and train a single policy ($\pi_\phi^g(a|s)$) to mimic teachers' behavior by minimizing KL divergence between them as follows:

$$\nabla_\phi \mathcal{L}_{RL} = \mathbb{E}_{g \sim \mathcal{U}} \left[ \mathbb{E}_{s \sim \pi_\phi^g} \left[ \nabla_\phi D_{KL} \left( \pi_T^g || \pi_\phi^g \right) + \alpha \nabla_\phi \mathcal{L}_{term} \right] \right] \tag{8}$$

where $D_{KL} \left( \pi_T^g || \pi_\phi^g \right) = \sum_a \pi_T^g(a|s) \log \frac{\pi_T^g(a|s)}{\pi_\phi^g(a|s)}$ and $\mathcal{U} \subset \mathcal{G}$ is the training set of subtasks. $\mathcal{L}_{term} = -\log \beta_\phi (s_t, g) = -\log P_\phi (s_t \in \mathcal{T}_g)$ is the cross-entropy loss for termination prediction. Intuitively, we sample a mini-batch of subtasks ($g$), use the subtask controller to generate episodes, and train it to predict teachers' actions. This method has been shown to be efficient for multi-task learning.

After policy distillation, the subtask controller is fine-tuned through actor-critic with generalized advantage estimation (GAE) (Schulman et al., 2015) as follows:

$$\nabla_\phi \mathcal{L}_{RL} = \mathbb{E}_{g \sim \mathcal{U}} \left[ \mathbb{E}_{s \sim \pi_\phi^g} \left[ -\nabla_\phi \log \pi_\phi (a_t|s_t, g) \hat{A}_t^{(\gamma, \lambda)} + \alpha \nabla_\phi \mathcal{L}_{term} \right] \right] \tag{9}$$

where $\hat{A}_t^{(\gamma, \lambda)} = \sum_{l=0}^\infty (\gamma \lambda)^l \delta_{t+l}^V$ and $\delta_t^V = r_t + \gamma V^\pi(s_{t+1}; \phi') - V^\pi(s_t; \phi')$. $\phi'$ is optimized to minimize $\mathbb{E} \left[ (R_t - V^\pi(s_t; \phi'))^2 \right]$. $\gamma, \lambda \in [0, 1]$ are a discount factor and a weight for balancing between bias and variance of the advantage estimation.

The final update rule for the subtask controller is:

$$\Delta \phi \propto - (\nabla_\phi \mathcal{L}_{RL} + \xi \nabla_\phi \mathcal{L}_{AM}) \tag{10}$$

where $\mathcal{L}_{AM} = \mathcal{L}_{sim} + \rho_1 \mathcal{L}_{dis} + \rho_2 \mathcal{L}_{diff}$ is the analogy-making regularizer defined as the weighted sum of three objectives described by Eq (4)-(6). $\rho_1, \rho_2, \xi$ are hyperparameters for each objective.

## E.2    META CONTROLLER

Actor-critic method with GAE is used to update the parameter of the meta controller. as follows:

$$\nabla_\theta \mathcal{L}_{RL} = - \begin{cases} \mathbb{E} \left[ c_t \left( \sum_i \nabla_\theta \log \pi_\theta \left( g_t^{(i)} | \mathbf{h}_t, \mathbf{r}_t \right) + \nabla_\theta \log P \left( \mathbf{l}_t | \mathbf{h}_t \right) \right) \hat{A}_t^{(\gamma, \lambda)} \right. & \text{(Hard)} \\ \left. + \nabla_\theta \log P \left( c_t | \mathbf{h}_t \right) \hat{A}_t^{(\gamma, \lambda)} + \eta \nabla_\theta \left\| \sigma \left( \varphi^{update} (\mathbf{h}_t) \right) \right\|_1 \right] \\ \mathbb{E} \left[ \sum_i \nabla_\theta \log \pi_\theta \left( g_t^{(i)} | \mathbf{h}_t, \mathbf{r}_t \right) \hat{A}_t^{(\gamma, \lambda)} \right] & \text{(Soft)} \end{cases} \tag{11}$$

where $c_t \sim P(c_t | \mathbf{h}_t) \propto \sigma \left( \varphi^{update} (\mathbf{h}_t) \right)$, and $P(\mathbf{l}_t | \mathbf{h}_t) \propto \text{Softmax} \left( \varphi^{shift} (\mathbf{h}_t) \right)$. $\eta$ is a weight for the update penalty.

The final update rule for the meta controller is:

$$\Delta \theta \propto - (\nabla_\theta \mathcal{L}_{RL} + \xi \nabla_\theta \mathcal{L}_{AM}) \tag{12}$$

where $\mathcal{L}_{AM}$ is the analogy-making regularizer. $\rho_1, \rho_2, \xi$ are hyperparameters for each objective.

# F   ARCHITECTURES AND HYPERPARAMETERS

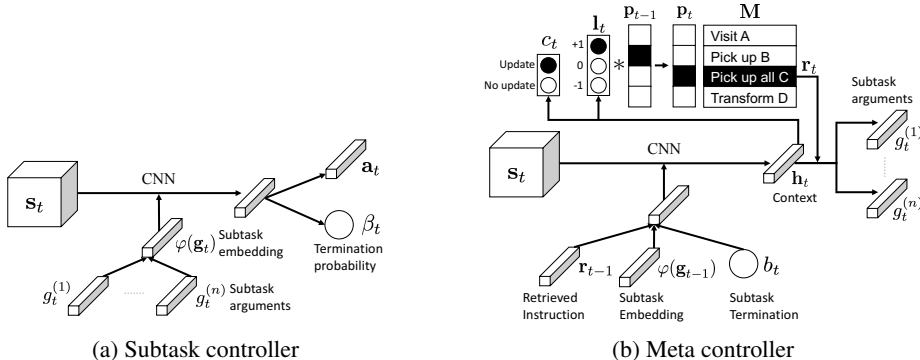

Figure 9: Proposed neural network architectures.

**Parameter Prediction.**    Parameter prediction approaches construct a neural network with parameters predicted by condition variables (e.g., exempler, class embedding). This approach has been shown to be effective for achieving zero-shot and one-shot generalization in image classification problems (Lei Ba et al., 2015; Bertinetto et al., 2016). More formally, given an input ($\mathbf{x}$), the output ($\mathbf{y}$) of a convolution and a fully-connected layer with parameters predicted by a condition variable ($\mathbf{g}$) can be written as:

$$\text{Convolution: } \mathbf{y} = \varphi\left(\mathbf{g}\right) * \mathbf{x} + \mathbf{b} \qquad \text{Fully-connected: } \mathbf{y} = \mathbf{W}' \text{diag}\left(\varphi\left(\mathbf{g}\right)\right)\mathbf{W}\mathbf{x} + \mathbf{b}$$

where $\varphi$ is the embedding of the condition variable learned by a multi-layer perceptron (MLP). Note that we use matrix factorization (similar to (Memisevic and Hinton, 2010)) to reduce the number of parameters for the fully-connected layer. Intuitively, the condition variable is converted to the weight of the convolution or fully-connected layer through multiplicative interactions. We used this approach as a building block to condition the policy network on the subtask embedding in the subtask controller and the meta controller.

**Subtask controller.**    The teacher architecture used for policy distillation is Conv1(32x3x3-1)-Pool(2)-Conv2(64x3x3-1)-FC1(256).[1] The network has two fully-connected output layers for actions and baseline respectively. The subtask controller architecture consists of Conv1(3x1x1-1)-Conv2(64x1x1-1)-Pool(2)-Conv3(128x3x3-1)-FC1(256) taking two recent observations as input. In addition, the subtask controller takes two subtask arguments ($g^{(1)}, g^{(2)}$) and computes ReLU($\mathbf{W}^{(1)}g^{(1)} \odot \mathbf{W}^{(2)}g^{(2)}$) to compute the subtask embedding. It is further linearly transformed into the weight of Conv1 and the (factorized) weight of FC1. Finally, the network has three fully-connected output layers for actions ($\varphi^{\pi}$), termination probability ($\varphi^{\beta}$), and baseline. In 'Concat' baseline architecture, the subtask embedding is linearly transformed and concatenated into the observation as 18 channels and FC1 as 256-dimensional vector.

We used RMSProp optimizer with the smoothing parameter of 0.97 and epsilon of $1e-6$. When training the teacher policy through actor-critic, we used a learning rate of $1e-3$. For training the subtask controller, we used a learning rate of $1e-3$ and $1e-4$ for policy distillation and actor-critic fine-tuning respectively. We used $\tau_{dis} = \tau_{diff} = 3, \alpha = 0.1$ for analogy-making regularization and the termination prediction objective. $\gamma = 0.99$ and $\lambda = 0.96$ are used as a discount factor and a balancing weight for GAE. 16 threads with batch size of 8 are used to run $16 \times 8$ episodes in parallel, and the parameter is updated after each run (1 iteration = $16 \times 8$ episodes). For better exploration, we applied entropy regularization with a weight of 0.01 and linearly decreased it to zero for the first 7500 iterations. The total number of iterations was 10K for both policy distillation and actor-critic fine-tuning.

**Meta Controller.**    The meta controller consists of Conv1(3x1x1-1)-Pool(2)-FC1(256) taking the current observation as input. The embedding of previously selected subtask ($\varphi(\mathbf{g}_{t-1})$), the previously retrieved instruction ($\mathbf{r}_{t-1}$), and the subtask termination ($b_t$) are concatenated and given as

---

[1] For convolution layers, NxKxK-P represents N kernels with size of KxK and padding of P. The number in Pool and FC represents the pooling size and the number of hidden units.

input for one-layer MLP to compute the joint embedding. This is further linearly transformed into the weight of Conv1 and FC1. The output of FC1 is used as the context vector ($\mathbf{h}_t$). We used the bag-of-words (BoW) representation as a sentence embedding which computes the sum of all word embeddings in a sentence: $\varphi^w(m_i) = \sum_{j=1}^{|m_i|} \mathbf{W}^m w_j$ where $\mathbf{W}^m$ is the word embedding matrix, each of which is 256-dimensional. An MLP with one hidden layer with 256 units is for $\varphi^{shift}$, a linear layer is used for $\varphi^{update}$. $\varphi^{goal}$ is an MLP with one hidden layer with 256 units that takes the concatenation of $\mathbf{r}_t$ and $\mathbf{h}_t$ as an input and computes the probability over subtask arguments as the outputs. The baseline network takes the concatenation of the memory pointer $\mathbf{p}_t$, a binary mask defined over memory locations indicating the presence of instruction, and the final hidden layer of $\varphi^{goal}$.

We used the same hyperparameters used in the subtask controller except that the batch size was 32 (1 iteration = $16 \times 32$ episodes). We trained the soft-architecture with a learning rate of $2.5e - 4$ using curriculum learning for 150K iterations, and fine-tune it with a learning rate of $1e - 4$ without curriculum learning for 25K iterations. Finally, we initialized the hard-architecture to the soft-architecture and fine-tune it using a learning rate of $1e - 4$ for 25K iterations. $\eta = 0.0001$ is used to penalize update decision.

**Flat Controller.** The flat controller architecture consists of Conv1(3x1x1-1)-Conv2(64x1x1-1)-Pool(2)-Conv3(128x3x3-1)-FC1(256) taking two recent observations as input. The previously retrieved instruction ($\mathbf{r}_{t-1}$) is transformed through an MLP with two hidden layers to compute the weight of Conv1 and FC1. The rest of the architecture is identical to the meta controller except that it does not learn temporal abstractions ($\varphi^{update}$) and has a softmax output over primitive actions.

**Curriculum Learning.** For training all architectures, we randomly sampled the size of the grid world from $\{7, 8, 9, 10\}$, the density of blocks and water cells are sampled from $[0, 0.1]$, and the density of objects are sampled from $[0, 0.6]$ for subtask pre-training, $[0, 0.15]$ for training on the easier environment, $[0, 0.3]$ for training on the original environment. We sampled the number of instructions from $\{1, 2, 3, 4\}$ for training the meta controller on the easier environment, but it was fixed to 4 for fine-tuning. The sampling range was determined based on the success rate of the agent.

