# Peer review of "Communicating Hierarchical Neural Controllers for Learning Zero-shot Task Generalization"

_ICLR 2017 — rejected_

[Official Review · AnonReviewer3 · rating 3 · confidence 4 · 16 Dec 2016]
**Review.**

The paper presents a hierarchical DRL algorithm that solves sequences of navigate-and-act tasks in a 2D maze domain. During training and evaluation, a list of sub-goals represented by text is given to the agent and its goal is to learn to use pre-learned skills in order to solve a list of sub-goals. The authors demonstrate that their method generalizes well to sequences of varying length as well as to new combinations of sub-goals (i.e., if the agent knows how to pick up a diamond and how to visit an apple, it can also visit the diamond). 

Overall, the paper is of high technical quality and presents an interesting and non-trivial combination of state-of-the-art advancements in Deep Learning (DL) and Deep Reinforcement Learning (DRL). In particular, the authors presents a DRL agent that is hierarchical in the sense that it can learn skills and plan using them. The skills are learned using a differential temporally extended memory networks with an attention mechanism. The authors also make a novel use of analogy making and parameter prediction. 

However, I find it difficult to understand from the paper why the presented problem is interesting and why hadn't it bee solved before. Since the domain being evaluated is a simple 2D maze, using deep networks is not well motivated. Similar problems have been solved using simpler models. In particular, there is a reach literature about planning with skills that had been ignored completely by the authors. Since all of the skills are trained prior to the evaluation of the hierarchical agent, the problem that is being solved is much more similar to supervised learning than reinforcement learning (since when using the pre-trained skills the reward is not particularly delayed). The generalization that is demonstrated seems to be limited to breaking a sentence (describing the subtask) into words (item, location, action). 

The paper is difficult to read, it is constantly switching between describing the algorithm and giving technical details. In particular, I find it to be overloaded with details that interfere with the general understanding of the paper. I suggest moving many of the implementation details into the appendix. The paper should be self-contained, please do not assume that the reader is familiar with all the methods that you use and introduce all the relevant notations. 

I believe that the paper will benefit from addressing the problems I described above and will make a better contribution to the community in a future conference.

[Official Review · AnonReviewer2 · rating 5 · confidence 5 · 17 Dec 2016]
**RL by learning to take advice**

This paper can be seen as instantiating a famous paper by the founder of AI John McCarthy on learning to take advice (which was studied in depth by other later researchers, such as Jack Mostow in the card game Hearts). The idea is that the agent is given high level instructions on how to solve a problem, and must distill from it a low level policy. This is quite related to how humans learn complex tasks in many domains (e.g., driving, where a driving instructor may provide advice such as "keep a certain distance from the car in front"). 

A fairly complex neural deep learning controller architecture is used, although the details of this system are somewhat confusing in terms of many details that are presented. A simpler approach might have been easier to follow, at least initially. The experiments unfortunately are on a rather simplistic 2D maze, and it would have been worthwhile to see how the approach scaled to more complex tasks of the sort usually seen in deep RL papers these days (e.g, Atari, physics simulators etc.). 

Nice overall idea, somewhat confusing description of the solution, and an inadequate set of experiments on a less than satisfactory domain of 2D grid worlds.

[Reviewer Comment · AnonReviewer4 · rating 7 · 02 Jan 2017]
**Novel architectural ideas ; algorithmically complex**

This paper presents an architecture and corresponding algorithms for
learning to act across multiple tasks, described in natural language.
The proposed system is hierarchical and is closely related to the options
framework. However, rather than learning a discrete set of options, it learns
a mapping from natural instructions to an embedding which implicitly (dynamically)
defines an option. This is a novel and interesting new perspective on options
which had only slightly been explored in the linear setting (see comments below).
I find the use of policy distillation particularly relevant for this setting.
This, on its own, could be a takeaway for many RL readers who might not necessarily
be interested about NLP applications.

In general, the paper does not describe a single, simple, end-to-end,
recipe for learning with this architecture. It rather relies on many recent
advances skillfully combined: generalized advantage estimation, analogy-making
regularizers, L1 regularization, memory addressing, matrix factorization,
policy distillation. I would have liked to see some analysis but
understand that it would have certainly been no easy task.
For example, when you say "while the parameters of the subtask controller are
frozen", this sounds to me like you're having some kind of two-timescale stochastic gradient
descent. I'm also unsure how you deal with the SMDP structure in your gradient
updates when you move to the "temporal abstractions" setting.

I am inclined to believe that this approach has the potential to scale up to
very large domains, but paper currently does not demonstrate this
empirically. Like any typical reviewer, I would be tempted to say that
you should perform larger experiments. However, I'm also glad that you have
shown that your system also performs well in a "toy" domain. The characterization
in figure 3 is insightful and makes a good point for the analogy regularizer
and need for hierarchy.

Overall, I think that the proposed architecture would inspire other researchers
and would be worth being presented at ICLR. It also contains novel elements
(subtask embeddings) which could be useful outside the deep and NLP communities
into the more "traditional" RL communities.

# Parameterized Options

Sutton et. al (1999) did not explore the concept
of *parameterized* options originally. It only came later, perhaps first with
["Optimal policy switching algorithms for reinforcement
learning, Comanici & Precup, 2010"] or
["Unified Inter and Intra Options Learning Using Policy Gradient Methods", Levy & Shimkin, 2011].
Konidaris also has a line of work  on "parametrized skills":
["Learning Parameterized Skills". da Silva, Konidaris, Barto, 2012)]
or ["Reinforcement Learning with Parameterized Actions". Masson, Ranchod, Konidaris, 2015].

Also, I feel that there is a very important distinction to be made with
the expression "parametrized options". In your work, "parametrized" comes in
two flavors. In the spirit of policy gradient methods,
we can have options whose policies and termination functions are represented
by function approximators (in the same way that we have function approximation
for value functions). Those options have parameters and we might call them
"parameterized" because of that. This is the setting of Comanicy & Precup (2010),
Levy & Shimkin (2011) Bacon & Precup (2015), Mankowitz, Mann, and
Mannor (2016) for example.

Now, there a second case where options/policies/skills take parameters *as inputs*
and act accordingly. This is what Konidaris & al. means by "parameterized", whose
meaning differs from the "function approximation" case above.
In your work, the embedding of subtasks arguments is the "input" to your options
and therefore behave as "parameters" in the sense of Konidaris.

# Related Work

I CTRL-F through the PDF but couldn't find references to any of S.R.K. Branavan's
work. Branavan's PhD thesis had to do with using control techniques from RL
in order to interpret natural instructions so as to achieve a goal. For example,
in "Reinforcement Learning for Mapping Instructions to Actions", an RL agent
learns from "Windows troubleshooting articles" to interact with UI elements
(environment) through a Softmax policy (over linear features) learned by policy
gradient methods.

As you mention under "Instruction execution" the focus of your work in
on generalization, which is not treated explicitely (afaik) in Branavan's work.
Still, it shares some important algorithmic and architectural similarities which
should be discussed explicitly or perhaps even compared to in your experiments
(as a baseline).

## Zero-shot and UVFA

It might also want to consider
"Learning Shared Representations for Value Functions in Multi-task
Reinforcement Learning", Borsa, Graepel, Shawe-Taylor]
under the section "zero-shot tasks generalization". 


# Minor Issues

I first read the abstract without knowing what the paper would be about
and got confused in the second sentence. You talk about "longer sequences of
previously seen instructions", but I didn't know what clearly
meant by "instructions" until the second to last sentence where you specify
"instructions described by *natural language*." You could perhaps
re-order the sentences to make it clear in the second sentence that you are
interested in NLP problems.

Zero-generalization: I was familiar with the term "one-shot" but not "zero-shot".
The way that the second sentence "[...] to have *similar* zero-shot [...]" follows
from the first sentence might as well hold for the "one-shot" setting. You
could perhaps add a citation to "zero-shot", or define it more
explicitly from the beginning and compare it to the one-shot setting. It could
also be useful if you explain how zero-shot relates to just the notion of
learning with "priors".

Under section 3, you say "cooperate with each other" which sounds to me very much
like a multi-agent setting, which your work does not explore in this way.
You might want to choose a different terminology or explain more precisely if there
is any connection with the multi-agent setting.

The second sentence of section 6 is way to long and difficult to parse. You could
probably split it in two or three sentences.

[Official Review · AnonReviewer1 · rating 4 · confidence 3 · 03 Jan 2017]
**Potentially good architecture; insufficient evaluation for "large-scale" tasks, no comparison to other state-of-the-art methods**

Description:

This paper presents a reinforcement learning architecture where, based on "natural-language" input, a meta-controller chooses subtasks and communicates them to a subtask controller that choose primitive actions, based on the communicated subtask. The goal is to scale up reinforcement learning agents to large-scale tasks.

The subtask controller embeds the subtask definition (arguments) into vectors by a multi-layer perceptron including an "analogy-making" regularization. The subtask vectors are combined with inputs at each layer of a CNN. CNN outputs (given the observation and the subtask) are then fed to one of two MLPs; one to compute action probabilities in the policy (exponential falloff of MLP outputs) and the other to compute termination probability (sigmoid from MLP outputs).

The meta controller takes a list of sentences as instructions embeds them into a sequence of subtask arguments (not necessarily a one-to-one mapping). A context vector is computed by a CNN from the observation, the previous sentence embedding, the previous subtask and its completion state. The subtask arguments are computed from the context vector through further mechanisms involving instruction retrieval from memory pointers, and hard/soft decisions whether to update the subtask or not.

Training involves policy distillation+actor-critic training for the subtask controller, and actor-critic training for the meta controller keeping the subtask controller frozen.

The system is tested in a grid world where the agent moves and interacts with (picks up/transforms) various item/enemy types.
It is compared to a) a flat controller not using a subtask controller, and b) subtask control by mere concatenation of the subtask embedding to the input with/without the analogy-making regularization.


Evaluation:

The proposed architecture seems reasonable, although it is not clear why the specific way of combining subtask embeddings in the subtask controller would be the "right" way to do it.

I do not feel the grid world here really represents a "large-scale task": in particular the 10x10 size of the grid is very small. This is disappointing since this was a main motivation of the work.

Moreover, the method is not compared to any state of the art alternatives. This is especially problematic because the test is not on established benchmarks. It is not really possible, based on the shown results, to put the performance in context of other works.

[Author Response · Junhyuk Oh · 18 Jan 2017]
**Common response to all reviewers**

Dear reviewers,

Thank you for your valuable comments. 
We have revised our paper by reflecting many comments from you.
We also added new results from a 3D visual domain to address concerns regarding simplicity of the 2D grid-world domain (Section 6.5).

We put a common response here as many of you raised similar questions/comments about complexity of our architecture and simplicity of the problem. We describe challenging aspects of our problem (not domain) and justify each component of our method. 
To begin with, the complex components of our architecture are designed NOT for the domain BUT for other challenges that we describe below: (1) zero-shot generalization over unseen tasks, (2) partial observability induced by instructions, and (3) mapping from instructions to subtasks. 

- Challenge 1:  Zero-shot generalization over unseen tasks
Most prior work on generalization in RL considers transfer learning where either the semantics of the task are fixed, or the agent is further trained on the target task. In contrast, our work considers “zero-shot” generalization where the agent should solve previously unseen tasks “without experiencing them beforehand”. In this setting, unlike conventional transfer learning, the agent needs to be given a description of the task (e.g., instructions) as additional input in order to be able to generalize to unseen tasks. Generalization over task descriptions is rarely tackled except for the papers we discussed in the related work section (e.g., UVFA [6], Parameterized Skill [3]). In this type of approach, it is necessary to learn a representation of task description (i.e., subtask in our work). We used a neural network to learn a representation of the subtask, and the term “subtask embedding” means such a learned representation. 

-- Why analogy-making regularization?
Simply learning the mapping from the subtask to the policy (or parameterized skill/option) does not guarantee that the learned mapping will generalize to unseen subtasks (i.e., “zero-shot” subtask generalization). This is why we proposed the analogy-making regularizer that allows the agent to learn the underlying manifold of the subtask space so that the agent can successfully map even an unseen subtask to a good policy. In our main experiment, we used analogy-making to encourage the agent to learn that it should perform any actions (e.g., pick up) on any target objects (e.g., cow) in the same way. Note that this is just one way of using our analogy-making regularizer. It can also address more complex generalization scenarios (e.g., “interact with X”) as discussed in Appendix B; in such cases, meaning of “interact” changes depending on the target objects, and simple methods (e.g., concatenation of action and target object embeddings) will fail in this scenario.

- Challenge 2: Partial observability induced by instructions
Much of the prior work on solving sequential RL tasks uses fully-observable environments [1, 2, 3, 5, 7] (i.e., a specific subtask to execute at each time step can be unambiguously inferred from the current observation). In contrast, our environment is “partially observable” because the agent is given just a full list of instructions, but it’s NOT given which instruction it has to execute at each time-step. In addition, the current observation (i.e., grid-world with objects) does not tell the agent which instruction to execute. Thus, the agent needs to “remember” how many instructions it has finished and decide when to move on to the next instruction. We chose this challenging setting motivated by the problem of a household robot that is required to execute a list of previously unseen instructions without human supervision that tells the robot what to do for every step. 

-- Why use memory?
We believe that our memory architecture is a much simplified version of Neural Turing Machines and has only the *minimal and necessary* components for dealing with partial observability described above. Without the memory component, there is no way for the agent to keep track of its progress on instruction execution. 

- Challenge 3: Mapping from instructions to subtasks 
Even though the meta controller is given a subtask controller that has pre-trained skills, the mapping from instructions to a sequence of skills (subtasks) is not trivial in our problem because of the following reasons: 1) stochastic events (i.e., randomly appearing enemy) require the agent to “interrupt” the current subtask. 2) “Transform/Pick up ALL” instructions require the agent to repeat the same subtask for a while, and the number of repetition depends on the observation. Moreover, the reward signal is quite delayed due to this type of instruction. 

-- Why differentiable temporal abstraction? 
In the meta controller, selecting a subtask at every time-step is inefficient and makes it harder to learn under delayed reward. It is known that “temporal abstraction” provided by options can allow the meta controller to learn faster as the meta controller can use the temporally-extended actions in SMDP [8]. However, the meta controller cannot simply use the subtask termination signal provided by the subtask controller to define the time-scale of its actions due to the necessity of “interruption” mechanism. Thus, we proposed a new way to “learn” the dynamic time-scale in the meta controller through neural networks. Although the agent can also deal with the challenges without such learned temporal abstraction in principle, we show empirically that the meta controller with our idea (learned temporal abstraction) performs significantly better than the baseline which updates the subtask at every time-step. 

- Justification of the use of other minor techniques
The three techniques listed below are existing methods that are applied to our problem in an appropriate way. Note that they are not designed to tackle the main challenges of our problem described above, but we used them to improve the overall results or stabilize training. We provided the reason why we used those techniques. 

-- Parameter prediction [9]: This approach has been shown to be effective for one-shot and zero-shot image classification problems. We used this technique because we also aim for zero-shot generalization over unseen tasks. 
-- Policy distillation [10]: This approach has been shown to be more efficient for multi-task policy learning. Although we found that policy distillation is not “necessary” to train the subtask controller, it gives slightly better results than training from scratch. 
-- Generalized Advantage Estimation [11]: This is a recent state-of-the-art technique to combine bootstrapped values and Monte-Carlo return to improve the stability and efficiency of training.

[References]
[1] George Konidaris, Andrew G. Barto. Building Portable Options: Skill Transfer in Reinforcement Learning, IJCAI 2007.
[2] George Konidaris, Ilya Scheidwasser, Andrew G. Barto. Transfer in Reinforcement Learning via Shared Features, Journal of Machine Learning Research 2012.
[3] Bruno Castro da Silva, George Konidaris, Andrew G. Barto, Learning Parameterized Skills, ICML 2012.
[4] Warwick Masson, George Konidaris, Reinforcement Learning with Parameterized Actions, AAAI 2016.
[5] S. R. K. Branavan, Harr Chen, Luke S. Zettlemoyer, Regina Barzilay, Reinforcement Learning for Mapping Instructions to Actions, ACL 2009.
[6] Tom Schaul, Daniel Horgan, Karol Gregor, David Silver. Universal Value Function Approximators, ICML 2015.
[7] Satinder Pal Singh. Transfer of learning by composing solutions of elemental sequential tasks. Machine Learning, 8(3-4):323–339, 1992.
[8] Richard S Sutton, Doina Precup, and Satinder Singh. Between mdps and semi-mdps: A framework for temporal abstraction in reinforcement learning. Artificial intelligence, 112(1):181–211, 1999.
[9] J. Lei Ba, K. Swersky, S. Fidler, et al. Predicting deep zero-shot convolutional neural networks using textual descriptions. CVPR 2015.
[10] A. A. Rusu, S. G. Colmenarejo, C. Gulcehre, G. Desjardins, J. Kirkpatrick, R. Pascanu, V. Mnih, K. Kavukcuoglu, and R. Hadsell. Policy distillation. ICLR 2016.
[11] J. Schulman, P. Moritz, S. Levine, M. Jordan, and P. Abbeel. High-dimensional continuous control using generalized advantage estimation. ICLR 2016.

[Final Decision · Program Chairs · 06 Feb 2017]
**ICLR committee final decision**

The paper looks at how natural language instructions can be decomposed into sub-tasks for as-yet-unseen new tasks
 hence the zero-shot generalization, which is considered to be the primary challenge to be solved. 
 The precise problem being solved by the original paper is not clearly expressed in the writing. This left some reviewers asking for comparisons, while the authors note that for the specific nature of the problem being solved, they could not seen any particular methods that is known to be capable of tackling this kind of problem. The complexity of the system and simplicity of the final examples were also found to be contradictory by a subset of the reviewers, although this is again related to the understanding of the problem being solved.
 
 With scores of 3/4/5/7, the ideas in this paper were appreciated by a subset of reviewers. 
 At the time of the writing of this metareview, the authors have posted a fairly lengthy rebuttal (Jan 18) and significant revisions (Jan 18), with no further responses from reviewers as of yet. However, it is difficult for reviewers to do a full re-evaluation of the paper on such short notice. 
 
 While the latest revisions are commendable, it is unfortunately difficult to argue strongly in favor of this paper at present.